# Effect of Fibre Orientation on Novel Continuous 3D-Printed Fibre-Reinforced Composites

**DOI:** 10.3390/polym13152524

**Published:** 2021-07-30

**Authors:** Ilaria Papa, Alessia Teresa Silvestri, Maria Rosaria Ricciardi, Valentina Lopresto, Antonino Squillace

**Affiliations:** 1Department of Chemical, Materials and Industrial Production Engineering, University of Naples Federico II, 80125 Naples, Italy; alessiateresa.silvestri@unina.it (A.T.S.); lopresto@unina.it (V.L.); squillac@unina.it (A.S.); 2Institute for Polymers, Composites and Biomaterials, CNR, 80078 Naples, Italy; mariarosaria.ricciardi@cnr.it; 3IMAST S.c.ar.l.—Technological District on Engineering of Polymeric and Composite Materials and Structures, Piazza Bovio 22, 80133 Napoli, Italy

**Keywords:** 3D printing, composites, Onyx, continuous fibre, mechanical characterisation, SEM, DSC, TGA

## Abstract

Among the several additive manufacturing techniques, fused filament fabrication (FFF) is a 3D printing technique that is fast, handy, and low cost, used to produce complex-shaped parts easily and quickly. FFF adds material layer by layer, saving energy, costs, raw material costs, and waste. Nevertheless, the mechanical properties of the thermoplastic materials involved are low compared to traditional engineering materials. This paper deals with the manufacturing of composite material laminates obtained by the Markforged continuous filament fabrication (CFF) technique, using an innovative matrix infilled by carbon nanofibre (Onyx), a high-strength thermoplastic material with an excellent surface finish and high resistance to chemical agents. Three macro-categories of samples were manufactured using Onyx and continuous carbon fibre to evaluate the effect of the fibre on mechanical features of the novel composites and their influence on surface finishes. SEM (Scanning Electron Microscopy) analysis and acquisition of roughness profile by a confocal lens were conducted. Tensile and compression tests, thermogravimetric analysis and calorimetric analysis using a DSC (differential scanning calorimeter) were carried out on all specimen types to evaluate the influence of the process parameters and layup configurations on the quality and mechanical behaviour of the 3D-printed samples.

## 1. Introduction

The growing interest in innovative and performing materials has made composites of interest in the research field due to their excellent strength/weight ratio and the multiplicity of application areas (aerospace, automotive, sports, and construction). As Antonucci [1] and Ricciardi [2] have shown, many studies are concerned with the traditional fabrication technologies of these materials, from protrusion to filament winding. They are obtained by different phases, such as polymer matrices and fibre reinforcements. On the other hand, other studies are concerned with optimising the processes for the existing techniques to get high-performance and, at the same time, light materials. However, as Ecker [3] demonstrated, these conventional techniques require moulds, raising the costs, and are limited to the performances reached up to now.

Brenken [4] has shown that additive manufacturing (AM) represents an evolution in this field, allowing the fabrication of composite materials in a polymeric matrix economically and flexibly. It works by processing layer by layer with the aid of a computer and a 3D design file.

An additive manufacturing process can obtain functional and even complex structures with high flexibility in fibre percentage and orientation, as Tran [5] demonstrates. Among the existing techniques, fused deposition modelling (FDM) or fused filament fabrication (FFF) is a fast, handy, and low-cost 3D printing technique mostly used to produce complex-shaped parts easily and quickly.

As in other AM technologies, FFF adds material layer by layer only where it is needed, saving energy, raw material costs, and waste. Moreover, the fabrication time can be reduced, especially when a complex geometry is required. However, despite its advantages over the conventional manufacturing process, FFF-moulded parts often present lower mechanical properties. First of all, the materials used for printing are mainly thermoplastic polymers, which are weaker than metals. Additionally, the fibre orientation leads to anisotropic material properties of the printed parts. The layer-by-layer print deposition process produces voids due to there being no good connection among the layers, resulting in lower tensile strengths. Therefore, it is necessary to improve the methods to improve the mechanical properties of the final product. This is precisely the concept taken up by Vanderploeg [6], stating that recently, the mechanical properties of the FFF filament have been improved by mixing the pure polymer with short carbon fibres. Carbon fibre mixed with the base thermoplastic polymer can significantly enhance the strength of the material and, therefore, can improve the properties of the FFF filament.

Recently, research has focused on developing polymer composites with continuous fibres using additive manufacturing technologies and developing new processes to enable the manufacture of parts with improved mechanical characteristics. Melnikova [7] describes some standard and novel materials used in the FFF technique and their influence on the resulting parts. Perkins [8] identifies 3D printing’s potential to enable buildings to be constructed many times faster and with significantly reduced labour costs. Wu [9], in a review, shows that 3D printing technology can be used to print large-scale structures.

However, the potential of the technology is limited by the lack of large-scale implementations, the development of building information modelling, and the life-cycle cost of the printed projects. Chiulan [10] and Liu [11] discuss the main 3D printing technologies currently employed in the case of poly (lactic acid) (PLA) and polyhydroxyalkanoates (PHA), two of the most important classes of thermoplastic aliphatic polyesters. Moreover, a short presentation of the main 3D printing methods is briefly discussed. Both PLA and PHA, in the form of filaments or powder, proved to be suitable for the fabrication of artificial tissue or scaffolds for bone regeneration. Tao [12] deals with the development of wood flour (WF)-filled polylactic acid (PLA) composite filaments for a fused deposition modelling (FDM) process with the aim of application to 3D printing. It demonstrates that compared with pure PLA filaments, adding WF changed the microstructure of the material fracture surface and the initial deformation resistance, and that the WF/PLA composite filament is suitable to be printed by the FDM process. Nguyen et al. [13,14] propose a route towards utilising lignin to replace petroleum-based thermoplastics (ABS) used in additive manufacturing and methods to enhance the printability of the materials with exceptional mechanical performance. They found that the presence of lignin and carbon fibres retards nylon crystallisation, leading to low-melting imperfect crystals. This allows good printability at lower temperatures without lignin degradation. Finally, T. Grimmelsmann [15] shows parameters affecting the interface layer. They show that both the printing material and the textile substrate influence the adhesion between both materials due to viscosity during printing, thickness, and pore sizes. While some material combinations build strong form-locking connections, others can easily be delaminated.

Several papers discuss conventional 3D-printed items [16,17], but very few works have been done on new advanced 3D-printed filaments. Wider requests of components manufactured by additive manufacturing are essential even if non-reinforced polymer filaments with low elastic and mechanical properties limit the applications. Therefore, the application of short fibre-reinforced filaments is encouraging. Few works address this issue [18,19]. It is important to characterise these printed samples for additional utilisation in advanced projects.

Among the infilled thermoplastic filaments used, only a few researchers, such as Mazzanti [20] and Sanei [21], deal with one of the most performing additive matrix (Onyx) polymers and the deposition by the new generation Markforged X7 3D printer. In particular, Sanei [21] shows that while the properties are lower than high-strength CFRPs (carbon-fibre-reinforced polymers) obtained by traditional technologies, there is a high potential for the use of 3D-printed composites to improve matrix and bonding properties between fibre and matrix.

Onyx is a commercial nylon mixed with short carbon fibre, and is a high-strength thermoplastic material with an excellent surface finish and high resistance to chemical agents. Onyx acts like a thermoplastic matrix for composite parts. It can be printed alone or reinforced with a continuous fibre to provide further resistance, and also gives the 3D-printed pieces a matte black surface finish. Instead of being customised from resin-impregnated carbon sheets, Onyx is manufactured from meticulously sliced fibres and combined with nylon, modifying the material’s performance on cooling, producing fewer thermal distortions. It is the best material for manufacturing components that necessitate great quality in agreement with engineering constraints. Onyx provides the strength of nylon with fibre hardness, high temperature endurance (T > 145 °C), and great endurance to unfavourable circumstances. The last manufacture does not need post-treatment thanks to a smooth and not shiny surface. If Onyx is used with fibres, the parts printed by Markforged printers are 30% stronger and harder than analogous components produced in other 3D printers. Unfortunately, the supplier of this innovative and high-performance material, such as Onyx, does not authorise scientific data to carry out CLT or FEM simulations that can guarantee better knowledge and predictions of the mechanical and printability performance of the components produced. For this reason, detailed production and test campaigns are necessary to have a global understanding of the material produced.

Based on what is said, the present paper deals with the manufacturing of composite material laminates obtained by the Markforged continuous filament fabrication (CFF) method using Onyx filaments as matrices and carbon for the fibre with different stacking sequences.

The production of the latter was separated into three macro-categories to evaluate the effect of the different configurations on the manufacturing and mechanical behaviour of the samples. The first one was made without using the reinforced fibre. In contrast, in the second and third configurations, the fibre was used in two different orientation layouts (0° and 0/90 with respect to the symmetry layer).

Each specimen was subjected to tensile and compression tests, thermogravimetric analysis, calorimetric analysis using DSC (differential scanning calorimeter), SEM (Scanning Electron Microscopy), and acquisition of roughness profile by a confocal lens. These tests revealed critical issues in the specimens regarding the adhesion of the fibre and the matrix as well as the formation of the voids, which drastically reduce the product’s mechanical properties. However, the performance of these reinforced materials can therefore be directly comparable to those of some high-performance and light materials such as aluminium alloys Al 7075 or AlSi10Mg.

## 2. Methodology

The current research activity deals with the manufacturing of 3D-printed composite material laminates realised by CFF, applying Onyx filaments and carbon fibre. The work aims to obtain more and more information about these materials and technology while not having the possibility, a priori, to carry out simulations on the behaviour of the components due to imposed limitations.

In order to obtain several additional pieces of information, the effect of different configurations on the manufacturing and mechanical behaviour of the items were evaluated through tensile and compression tests, thermogravimetric analysis, calorimetric analysis using DSC (differential scanning calorimeter), SEM (Scanning Electron Microscopy), and acquisition of roughness profile by a confocal lens.

### Materials and Manufacturing

The 3D-printed composite samples were produced using two commercial products by Markforged: Onyx and continuous carbon fibre. Onyx is made of nylon blended with short carbon fibre and represents a high-resistance thermoplastic material, with an excellent surface finish and high resistance to chemical agents. Onyx acts like a thermoplastic matrix for composite parts and can be printed on its own or reinforced with a continuous fibre for added strength, and also gives 3D-printed parts a matte black surface finish. Compared to traditional nylon, Onyx is about 3.5 times stronger. In addition, it has a higher hardness and a HDT (heat deflection temperature) of 140 °C. Infilled short carbon fibres in nylon modify the behavior of the cooling material, inducing fewer thermal deformations so that the dimensions of the printed pieces faithfully reproduce the model designed in CAD. Given the excellent surface finish obtained, it is assumed that no further processing is required. Material properties for Onyx reported by the manufacturer are shown in Table 1 [6].

The long carbon fibre (FRP or fibre-reinforced polymer) is made up of thin carbon filaments. Carbon fibre filament has extremely variable chemical and physical properties, and it is a material that has a filamentous structure [22]. Each twine of carbon filaments is formed by the union of many thousands of fibres. Each filament has an approximately cylindrical shape with a diameter of 5–8 μm, and consists almost exclusively of carbon. It is a material characterised by high mechanical strength, low density, good thermal insulation capacity, resistance to temperature changes and the effects of chemical agents, and good flame retardant properties (see Table 2).

The modulus of elasticity varies from about 350 GPa, a value between the glass fibres and aluminium ones, up to 7000 GPa, or three times greater than that of steel, as easily comparable from Silvestri’s work [23].

The printing machine chosen is the Markforged X7, which allows reinforcing parts with fibre deposits to achieve an unmatched strength. Thanks to these 3D printer characteristics, robust and resistant samples were printed, and the process’s repeatability is assessed.

The entire process begins with the part’s design to be printed in a CAD model, and then uploading an STL file directly to the software associated with the machine. Then, by selecting printing parameters such as materials, deposition strategy, and more, the cloud-based software does the remainder by quickly printing the correct part.

In other words, there are two independent extruders, one dedicated to the plastic filament and one dedicated to the fibre filament. The extruders can generate a composite part, one layer at a time according to the continuous filament fabrication (CFF) process technology, on the deposition plate. The first nozzle builds the plastic matrix and the second wraps the fibre, not working simultaneously, but one by one as suggested by the selected configuration. The extruders move along x and y directions while the print bed moves along the z direction. In Figure 1, a scheme of CFF technology for the 3D-printed polymer-based materials is shown.

With a built-in laser micrometre, the printer automatically scans the processing plate with a precision of 1 um to create a contour map of its surface. At this point, it calibrates its measurements with the extrusion readings to correctly set the height of the nozzle and the dynamic adjustment of the topography. This way, the printer ensures that the printouts are exactly as designed.

Each sample is manufactured with two top and bottom Onyx layers depositing the matrix with 100% infill following a 45-degree pattern with extrusion temperature T = 220 °C and deposition speed 6.90 cm^3^/h. A drawing of the matrix scan pattern applied is reported in Figure 2.

Three macro-typologies of specimens were created and referred to below as FO (full Onyx), 0FC (zero fibre content), and 090FC (090 fibre content). For 100% filler (FO), all the Onyx layers are deposited as the top and bottom ones above are. The reinforced configurations (0FC and 090FC) are characterised by the same fibre volume fractions Vf = 69% and different layup sequences (Figure 3). Each layer of the stratified panel is about 0.125 mm in thickness, with a final thickness of the samples equal to t = 2 ± 0.2 mm. In 0FC composites, the fibre is deposited at 0° with respect to each layer’s plate. In 090FC samples, the fibre is deposited alternatively at 0 and 90 degrees. A deposition speed of 2.39 cm^3^/h for the carbon fibre layers was applied for both configurations. Five samples of each specimen configuration were manufactured.

The standard test method used in this study follows the specification from ASTM D3410 for compression tests and ASTM D3039 for tensile tests (Table 3).

The thermogravimetric analysis (TGA) was carried out by TGA Q5000, capable of reaching a temperature of 1000 °C with a maximum speed of 100 °C/min. The analysis was conducted on the two selected samples to identify the thermal degradation intervals of the material under examination and to obtain information relating to weight losses. A thermal program with a constant temperature increase of 10 °C/min was used to scan tests from 30 °C to 600 °C. The samples used were about 10 mg, considering that the materials used are fibre reinforced and carbon fibres.

The calorimetric analysis (DSC) was conducted with a differential scanning calorimeter from TA Instruments (TRIOS). The tests were carried out on both carbon fibre matrix samples and PA6 matrix alone to evaluate the effect of the presence of the semi-crystalline polymer on the melting and crystallisation of the system. All items underwent a double heating scan from −50 °C to 250 °C, with a heating rate of 10 °C/min in a nitrogen atmosphere and a cooling scan from 250 °C to −50 °C with the same speed of 10 °C/min. The double heat scan was performed to give the various samples an equivalent thermal history. The capsules for the DSC were filled with an average of 10–15 mg of material.

The surface finish of each manufactured panel was observed using a confocal Leica DCM3D scan and optical microscopes, with the aim being to analyse the surface pattern roughness.

Moreover, to investigate the mechanical properties of the samples, uniaxial tensile and compression tests were carried out by using a Galdabini QUASAR 50 testing machine. The latter is equipped with a 50 kN load cell and a micron extensometer to measure the deformation up to the rupture; tests were carried out at room temperature and with a testing speed set to 3 mm/min. According to international standards, the software associated with the Galdabini machine, LabTest, allows for programming the tests.

Finally, to analyse the quality of the printing process and better understand the phenomena involving the fracture of the samples, SEM analysis was carried out using the Hitachi TM3000 scanning electron microscope.

## 3. Results and Discussion

### 3.1. TGA and DSC

Thermogravimetry is a beneficial technique because it allows for the evaluation of the thermal stability of polymeric materials, especially the possibility of use at temperatures higher than that of the environment. Thermal resistance is given by the maximum temperature to which materials can be heated without undergoing irreversible chemical changes, corresponding to their physicochemical properties. During thermogravimetry, the formation of volatile compounds is proof of the occurrence of an irreversible chemical process: thermal degradation.

The thermogravimetric analysis was carried out to verify the maximum temperature used in the calorimetric study. Thus, it has a more global view of the operating temperatures for 3D printing.

Thermogravimetric tests have shown that in the temperature range between 400 and 500 °C, Onyx has a greater weight loss than carbon fibres alone, of about 8%. This result, seeing the final residue of the composite with respect to the fibre alone, leads to the conclusion that 30% of this material for 3D printing is made up of a matrix and therefore to find the optimal process parameters to achieve the most suitable configuration it is necessary to carry out a calorimetric analysis to verify possible transitions.

Below, the thermal properties of the two samples in an inert environment are provided (Table 4).

Finally, Figure 4a,b shows the TGA results for Onyx and carbon fibre, respectively. The TGA analysis has been carried out up to T = 205 °C since the Markforged 3D printer used is a limited machine and requires printing in a temperature range not exceeding 250 °C for the selected material. In Figure 4b it is possible to observe how some melting or crystallisation peaks are linked to the presence of plasticisers in the carbon fibres, which do not seem to make any useful contribution to the properties of the composite. At T = 200 °C, a mass loss of about 2% is recorded for the carbon fibre (Figure 4b). However, this minimal and negligible weight loss is to be attributed to the size of the fibre compatible with the matrix used, which is helpful in promoting adhesion between the layers.

In Table 5, thermal transactions of the analysed material and the result of the DSC analysis for the Onyx samples are reported. The first heating, first cooling, and second heating scan of an Onyx multifilament at 10 °C/min are given in Figure 5. The heating scans show the presence of a maximum of two melting peaks. Cooling of the Onyx by DSC results in one crystallisation peak. The melting peak around 192 °C could be attributed to the melting of the α-crystalline form. In the second heating scan, the small endothermic peak indicates the new organisation and crystallisation during the DSC scan. The crystallisation peak extends from 145 °C to 180 °C, and the peak temperature is about 166 °C. Since the fusion temperature is 192 °C, it was decided to use a temperature of 215 °C as a process parameter for the extrusion of the Onyx, slightly higher than the melting temperature. However, from analysis of this material at this temperature, there appears to be shallow weight loss and no thermal degradation of the material.

### 3.2. Roughness Evaluation

Table 6 shows the average results obtained from the measurement of the surface roughness of the three types of specimens. Thanks to the deposition of two skins (surface and top layer) valid for the surface finish, the roughness values measured on the reinforced samples are not similar. As for the importance of the parameters obtained, R_a_ = 13.65 µm does not denote an excellent surface finish. Still, the dependence of the latter on the materials used and the printing parameters must certainly be thoroughly assessed. It should be remembered that R_a_ is the average roughness value extended to the surface and has to be taken into account, given the anisotropy of the components to be analysed.

### 3.3. Results from Tensile and Compression Tests

The results of both mechanical tests are shown in Figure 6, Figure 7 and Figure 8. In addition, the tensile modulus of elasticity (E), tensile load (Ftu), and compression (Fcu) strength are reported, respectively.

Regarding the maximum tensile load value for specimens 0FC and 090FC, it can be seen that compared to specimen FO, a considerable improvement has been obtained, mainly due to the properties of the carbon fibre reinforcement (Ftu = 700 MPa, E = 54 GPa). In particular, in the tensile test, specimen 0FC, with unidirectional reinforcement, reaches the value closest to that of the carbon fibre (Ftu = 581 MPa, E = 62.5 GPa) and almost double that of the 090FC one (Ftu = 324 MPa, E = 36.4 GPa).

It is not a coincidence that the performance of these reinforced materials can therefore be directly comparable to those of some high-performance and light materials, such as aluminium alloys Al 7075 or AlSi10Mgas, as reported by Silvestri [17]. Therefore, it is clear what impact the very strong push that research has had in understanding and governing the new production paradigms imposed by additive manufacturing.

The obtained results confirm the highly anisotropic behaviour of the material, which, for 0FC samples, has values of tensile resistance and modulus of elasticity considerably higher than those of 090FC, respectively by 44.2% and 77.2%.

On the other hand, in the compression tests the presence of orthogonal reinforcing fibres in 090FC allowed the acquisition of significantly better results (Fcu = 75 MPa, E = 7.2 GPa) than for the 0FC ones (Fcu = 62 MPa, E = 6.5 GPa).

In general, after fibre fracture, ductile behaviour was shown with significant plastic deformation in the case of the FO samples. However, this does not happen in reinforced samples where the more brittle break indicates a poor adhesion to the matrix fibre (Figure 9).

### 3.4. SEM Fracture Analysis Surface

Optical SEM microscopy was performed to assess the quality of the printed samples further. Figure 10 shows the acquisition of the fracture surfaces of the samples tested.

Starting from the FO specimen (Figure 10a), it can be seen that the printing process is characterised by a high degree of porosity and voids.

Observing the reinforced 0FC samples (Figure 10b), a degree of porosity and the pull-out phenomenon can also be seen here, or when there is an incorrect load transfer from matrix to fibre, probably due to poor impregnation of the fibre. This leads the fibres to slip out of the matrix, still intact, without participating in the material’s resistance. Finally, in the 090FC-type specimens (Figure 10c), it is possible to appreciate a certain degree of misalignment of the fibres due to a non-optimal fibre-matrix adhesion.

It should also be noted that the matrix has a considerable elongation at break, a phenomenon typical of nylon matrices.

A 7–11% void content was estimated from the micrograph using CAD software and a greyscale threshold.

## 4. Conclusions

In this work, specimens of composite material printed in additive manufacturing were produced using the continuous filament fabrication technology with a double extruder proposed by Markforged.

The material chosen for the matrix is Onyx, a thermoplastic composed of nylon mixed with short carbon fibre. For the reinforcement, the material chosen is carbon fibre.

Therefore, three types of specimens were produced only in Onyx, with carbon fibre reinforcement oriented at 0° and finally alternating the fibres at 0° and 90° up to the symmetry layer.

From the mechanical tests carried out on the three configurations, it was deduced that:0° reinforcement specimens achieved a significant improvement in tensile strength over matrix-only specimens0°/90° reinforcement specimens did not achieve significant tensile strength improvements over matrix-only specimens

However, the performance of these reinforced materials can therefore be directly comparable to those of some high-performance and light materials, such as aluminium alloys Al 7075 or AlSi10Mg.

After carrying out the load tests, further tests were carried out to characterise the material from a chemical-physical point of view, allowing the observation of some typical defects of CFF processes such as voids and porosity and non-optimal fibre-matrix adhesion.

The results of the topographic characterisation confirmed that the degree of finish of the printed specimens is not optimal but strongly influenced by the materials and printing parameters.

Further investigation is necessary to determine the characteristics of the composite. In particular, multiaxial load tests should be carried out to evaluate the properties in all directions, given the strong anisotropy of the items produced with these technologies, or impact tests, in which composite materials are usually deficient.

Further studies are also required to fully understand and govern the adhesion and impregnation processes of the fibre within the composite. Voids and porosity can be reduced by intervening both in the 3D printing parameters (printing speed, extruder temperature) and subjecting the product to hot compression after moulding.

## Figures and Tables

**Figure 1 polymers-13-02524-f001:**
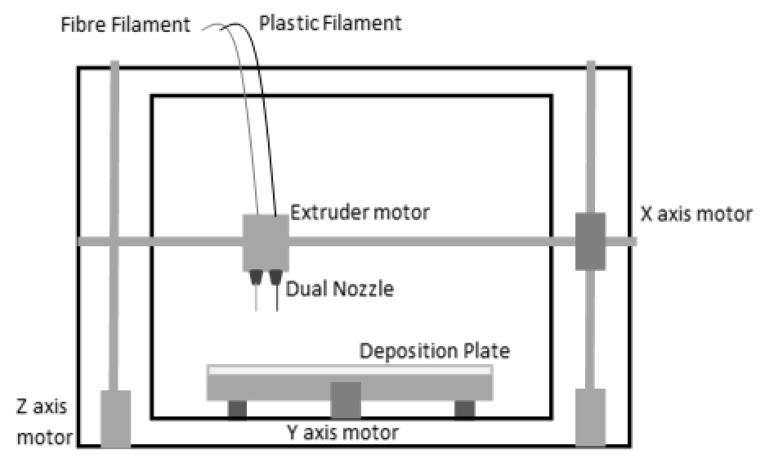
Markforged dual extruder deposition system.

**Figure 2 polymers-13-02524-f002:**
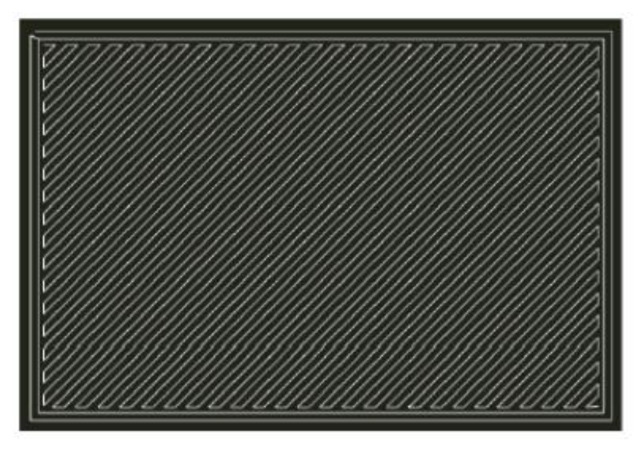
Direction of deposition of the individual layers; top and bottom layers and FO samples.

**Figure 3 polymers-13-02524-f003:**
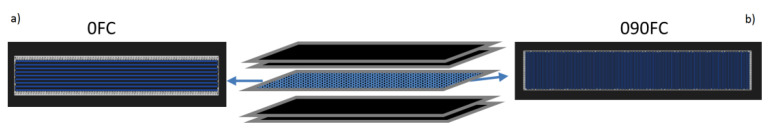
Scheme of fibre stacking sequences: (**a**) 0FC; (**b**) 090FC.

**Figure 4 polymers-13-02524-f004:**
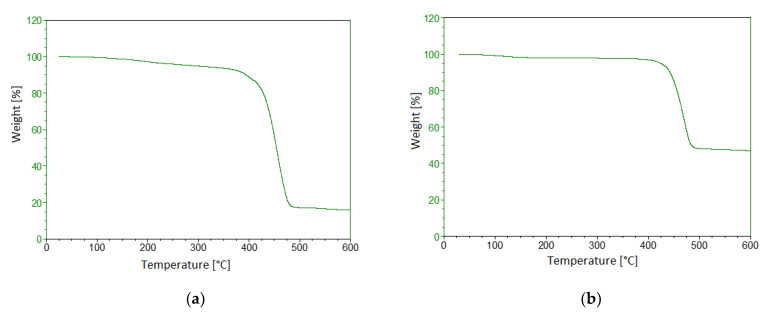
TGA results. (**a**) Onyx; (**b**) carbon fibre.

**Figure 5 polymers-13-02524-f005:**
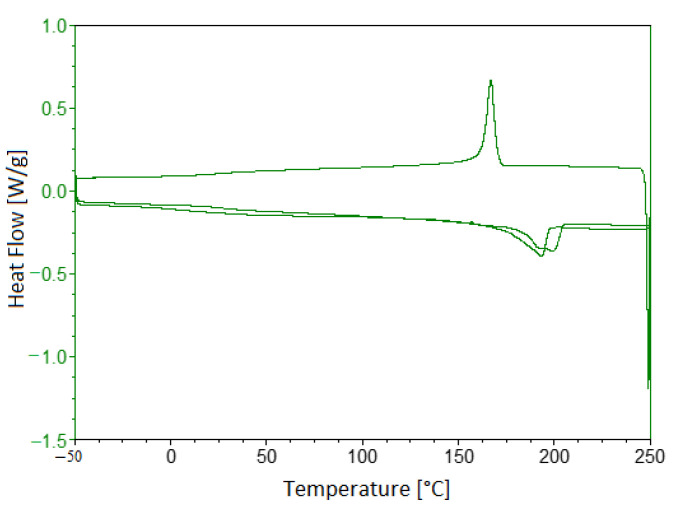
DSC results for Onyx sample.

**Figure 6 polymers-13-02524-f006:**
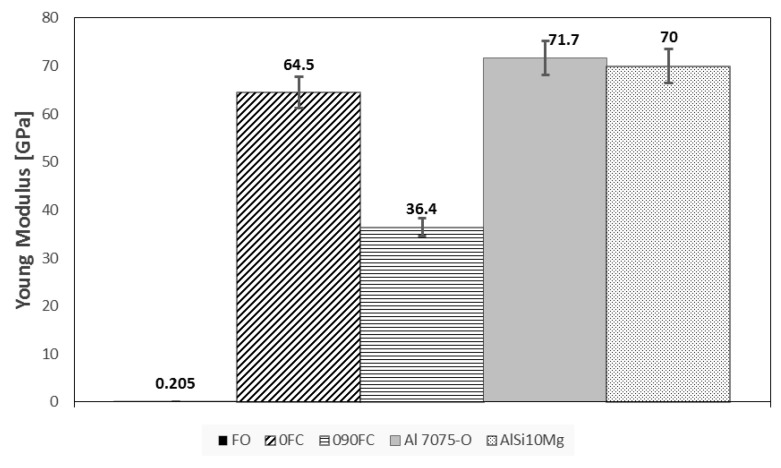
Tensile Young’s modulus for all systems.

**Figure 7 polymers-13-02524-f007:**
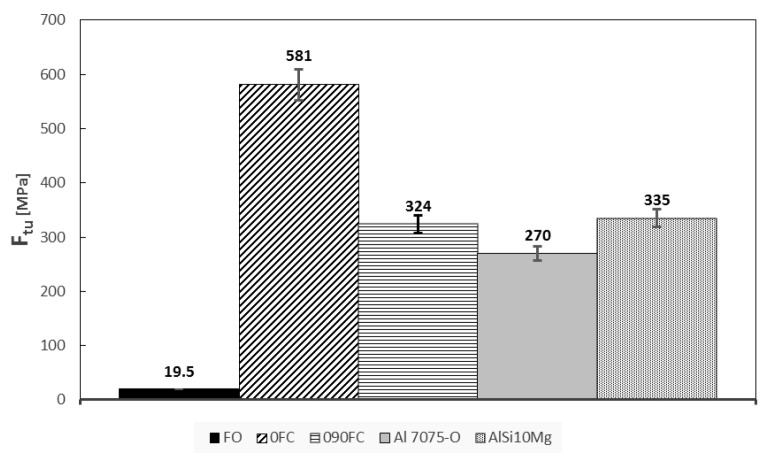
Final tensile strength for all systems.

**Figure 8 polymers-13-02524-f008:**
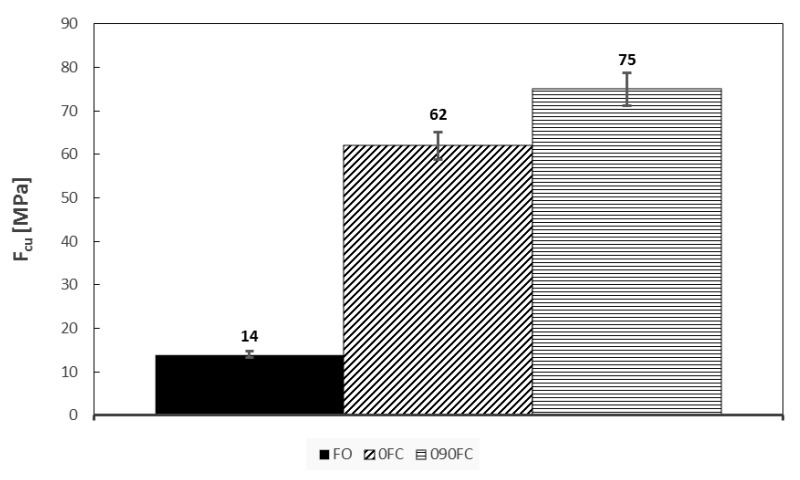
Final compression strength for all systems.

**Figure 9 polymers-13-02524-f009:**
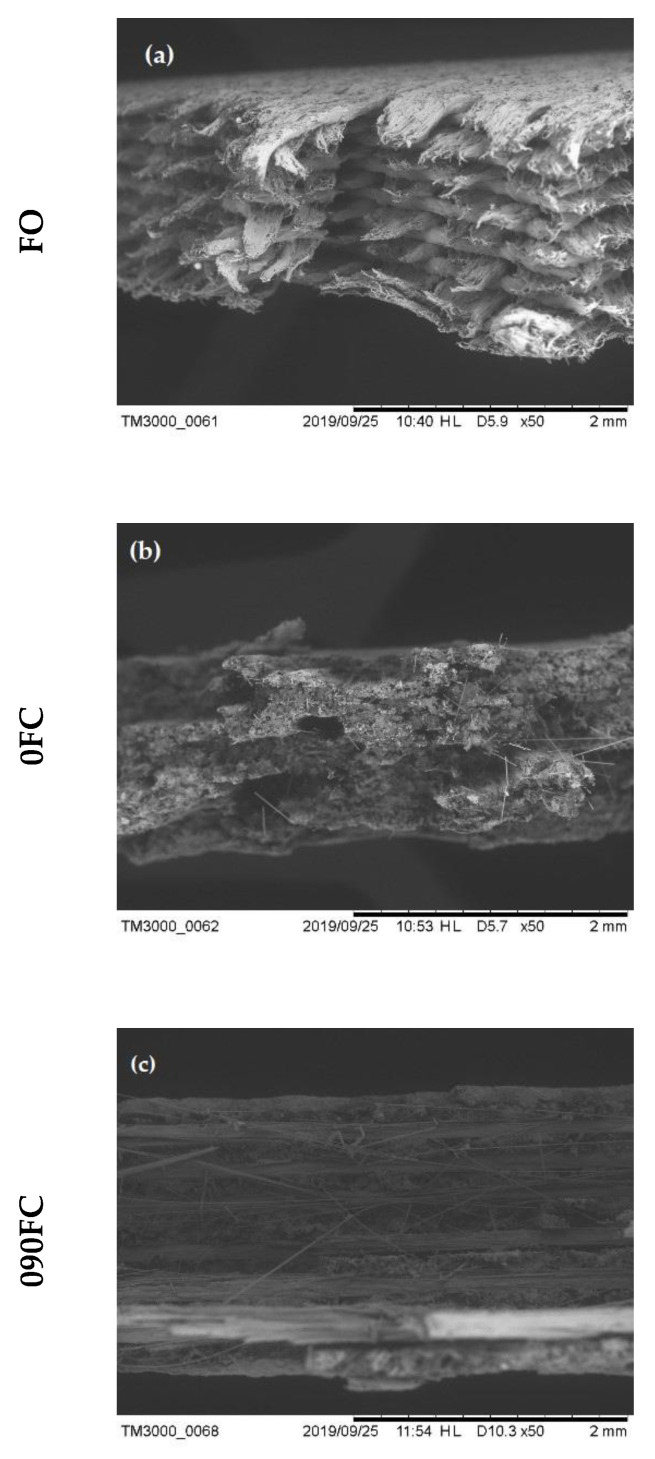
Pictures of tensile break mode of samples: (**a**) FO; (**b**) 0FC; and (**c**) 090FC.

**Figure 10 polymers-13-02524-f010:**
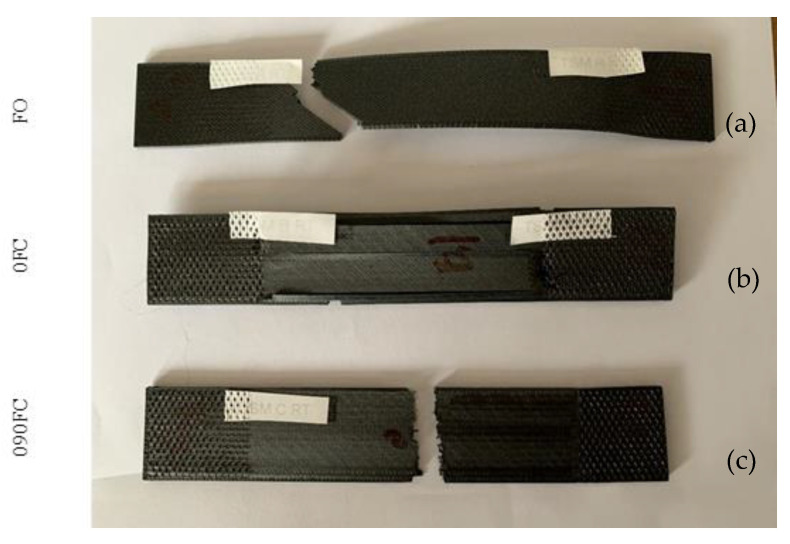
SEM analysis of the fracture surface: (**a**) FO; (**b**) 0FC; and (**c**) 090FC.

**Table 1 polymers-13-02524-t001:** Onyx mechanical features.

Material	Density (g/cm^3^)	Filament Diameter (μm)	Tensile Modulus (GPa)	Flexural Modulus (GPa)
Onyx	1.2	1750	1.4	2.9

**Table 2 polymers-13-02524-t002:** Carbon fibre filament properties.

Fibre Reinforcement	Test (ASTM)	Carbon
Tensile Strength (Mpa)	D3039	800
Tensile Modulus (GPa)	D3039	60
Tensile Strain at Break (%)	D3039	1.5
Flexural Strength (Mpa)	D790	540
Flexursl Modulus (GPa)	D790	51
Flexural Strain at Break (%)	D790	1.2
Compressive Strength (MPa)	D6641	320
Compressive Modulus (MPa)	D6641	54
Compressive Strain at Break (%)	D6641	0.7
Heat Deflection Temp (°C)	D648 B	105
Izod Impact—Notched (J/m)	D256-10 A	960
Density (g/cm^3^)	--	1.4

**Table 3 polymers-13-02524-t003:** Specimen dimensions.

Test Type	Width, t (mm)	Length, L (mm)
Tensile	2.2	24.9
Compression	2.1	9.9

**Table 4 polymers-13-02524-t004:** TGA results.

Sample	Temperature (°C)	Weight Loss %	Degradation Range (°C)	Weight Loss %	The Residue %
Onyx	400	11	400–500	82.77	16
CF	400	3	400–500	52	47

**Table 5 polymers-13-02524-t005:** DSC results.

Sample	T_m_ (°C)	ΔH_m_ (J/g)	Tc (°C)	ΔHc (J/g)
Onyx	192.96	32.08	166.93	37.05

**Table 6 polymers-13-02524-t006:** Roughness parameters for all samples manufactured.

	A	B	C	R_a_m_
S_a_ [µm]	13.0 ± 0.5	14.1 ± 0.5	15.3 ± 0.5	13.65
S_z_ [µm]	405 ± 10	394 ± 8	557 ± 12	435.5
S_sk_	−0.16 ± 0.04	−0.255 ± 0.03	0.146 ± 0.02	−0.08
W_sm_ [mm]	0.558 ± 0.1	0.410 ± 0.1	0.481 ± 0.1	0.474
W_a_ [µm]	5.88 ± 1.0	6.65 ± 2	15.3 ± 1	6.215

## Data Availability

Not applicable.

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
