# Peer review of "Effect of Fibre Orientation on Novel Continuous 3D-Printed Fibre-Reinforced Composites"

_polymers, 2021, doi:10.3390/polym13152524_

Round 1

Reviewer 1 Report

The results of this study can contribute certain insights to the field of additive manufacturing. The manuscript can be accepted to publish in Polymers after minor revision.

The author should check some spelling of the manuscript, for example page 2, line 74.

The introduction should have some additional details and references related to chopped carbon fiber and continuous carbon fiber composites.

The discussion of using nylon- chopped carbon fiber composites, viscosity, and thermal characteristics or interlayer bonding of the 3D-printed objects (lines 80-92, page 2) can be reviewed by:

Applied Materials Today 12 (2018): 138-152.

Science advances 4, no. 12 (2018): eaat4967.

The authors should discuss degradation temperature of the composite at 300 C. All DSC measurements were performed from -50 to 300C. It looks like that there is around 5% weight loss at 300C for the composite. The DSC measurements could be performed at temperature up to 250 C, low weight loss (melting temperature of nylon is from 150- 210C).

The TGA data (Fig. 4b) showed that there is a weight loss at around 100- 200C of carbon fiber, what is it? The author should discuss about it.

Page 8, line 273:  The recrystallization temperature of sample is from about 145 to 180 C, not 170-230 C, please correct it.

Author Response

The results of this study can contribute certain insights to the field of additive manufacturing. The manuscript can be accepted to publish in Polymers after minor revision.

1Q. The author should check some spelling of the manuscript, for example page 2, line 74.

1A. Thanks. The sentence has been modified, and a complete check of the text has been done.

2Q. The introduction should have some additional details and references related to chopped carbon fiber and continuous carbon fiber composites

2A. Thanks for the revision. Additional details and references about chopped and continuous fibre have been added in the introduction. The references are reported below

  1. Ahn, S.,Montero, M., Odell, D.,Roundy S., Wright, P. K. Anisotropic material properties of fused deposition modeling ABS’, Rapid Prototyping J.,
  2. Melenka, G. W., Schofield, J. S., Dawson, M.R., Carey J. P. Evaluation of dimensional accuracy and material properties of the MakerBot 3D desktop printer, Rapid Protot J.,
  3. Ning, F., Cong, W., Qiu, J., Wei J., Wang, S. Additive manufacturing of carbon fiber reinforced thermoplastic composites using fused deposition modeling, Compos Part B: Eng., 2015.
  4. Ivey, M., Melenka, G.W. Jason, Carey P., Ayranci, C. Characterizing short-fiber-reinforced composites produced using additive manufacturing , Manufact. Pol. & Comp. Sc., 2017

3Q.The discussion of using nylon- chopped carbon fiber composites, viscosity, and thermal characteristics or interlayer bonding of the 3D-printed objects (lines 80-92, page 2) can be reviewed by:

Applied Materials Today 12 (2018): 138-152.

Science advances 4, no. 12 (2018): eaat4967.

3A. Thanks, the references have been added and discussed.

4Q. The authors should discuss degradation temperature of the composite at 300 C. All DSC measurements were performed from -50 to 300C. It looks like that there is around 5% weight loss at 300C for the composite. The DSC measurements could be performed at temperature up to 250 C, low weight loss (melting temperature of nylon is from 150- 210C).

4A. Thanks for the revision. As seen from the TGAs below, a weight loss of 2% and 4% is recorded for the matrix and fiber. However, it was not our interest to go beyond a T = 250 ° C despite the minimal weight loss as the Markforged 3d printer used is a limited machine. Therefore, it requires printing in a temperature range not exceeding 250 ° C for the selected material.

.

Figure 1. TGA Onyx

Figure 2. TGA Carbon fibre, CF

5Q. The TGA data (Fig. 4b) showed that there is a weight loss at around 100- 200C of carbon fiber, what is it? The author should discuss about it.

5A. As can be seen from Figure 2 above, at T = 200 ° C a mass loss of about 2% is recorded for the carbon fiber alone. Indeed, this minimal and negligible weight loss is to be attributed to the size of the fiber compatible with the matrix used and helpful in promoting adhesion between the layers. However, the discussion of Figure 4b has been upgraded.

6Q. Page 8, line 273:  The recrystallization temperature of sample is from about 145 to 180 C, not 170-230 C, please correct it.

6A. Thanks, the temperatures have been modified.

Reviewer 2 Report

The manuscript deals with the evaluation of fiber orientation of continuos 3D printing novel material

In section 2.1, it is not clear of the onyx material is produced in the laboratory or is commercial material. In any case, authors need to add more concise information. The same should be applied to the CF.

Also, it is not clear how the authors place de CF in the laminated 3D Printing, please write a paragraph that explains this. I want to understand if the CF is a printed or separately layer, if it is printed, how the authors made this through 3D printing, it is totally confusing.

Apendix A and B? please verify

Author Response

The manuscript deals with the evaluation of fiber orientation of continuos 3D printing novel material

Q1. In section 2.1, it is not clear of the onyx material is produced in the laboratory or is commercial material. In any case, authors need to add more concise information. The same should be applied to the CF.

A1. Thanks. Both Onyx and the carbon fiber used are two commercial materials products by Markforged. The information has been added in both Introduction and section 2.1. In addition, a reference for the CF has been added to have more information about the material.

Kabir, S.M.F., Mathur, K., Seyam, A.F.M. A critical review on 3D printed continuous fiber-reinforced composites: History, mechanism, materials and properties. Compos Struct; 2020.

Q2. Also, it is not clear how the authors place de CF in the laminated 3D Printing, please write a paragraph that explains this. I want to understand if the CF is a printed or separately layer, if it is printed, how the authors made this through 3D printing, it is totally confusing.

A2. Thanks for the suggestion. The 3d printing machine has two extruders independent, one dedicated to the plastic filament and one dedicated to the fibre filament. The extruders can generate a composite part one layer at a time according to the Continuous Fiber Fabrication (CFF) process technology on the deposition plate. So, the first nozzle builds the plastic matrix, and the second wrap the fibre not working simultaneously, but one by one as suggested by the selected configuration. The extruders move along x- and y- directions while the print bed moves along the z-direction. In figure 1, a scheme of CFF technology for the 3D-printed polymer-based materials is shown. The information has been added in the text.

Q3. Apendix A and B? please verify

A3. Thanks for the revision. The sentences have been deleted.

Reviewer 3 Report

This is an excellent written and discussed paper. We'll done to authors. I would like to see a short paragraph related to the direct implementation of this very interesting approach 

Author Response

Thanks for the positive review. Unfortunately, it is not possible to provide more details on the innovative implementation covered in the paper. The carbon fibre-filled thermoplastic matrix used is a proprietary commercial product from Markforged. This manufacturer prevents the detailed knowledge of the material mentioned above (ONYX), preventing numerical simulations from deepening their understanding. For this reason, an accurate and complete characterization of the material is necessary to evaluate its performance and try to improve its performance.
